# Correlation of preoperative frailty with postoperative delirium and one-year mortality in Chinese geriatric patients undergoing noncardiac surgery: Study protocol for a prospective observational cohort study

**Min Zhang[1☯], Xiaojun Gao[1☯], Mengjie Liu[1], Zhongquan Gao[1,2], Xiaxuan Sun[1,2], Linlin Huang[3], Ting Zou[3], Yongle Guo[1,2], Lina Chen[1], Yang Liu[1], Xiaoning Zhang[1], Hai Feng[1], Yuelan Wang[4], Yongtao Sun [1] ***

1 Department of Anesthesiology, Shandong Institute of Anesthesia and Respiratory Critical Medicine, The First Affiliated Hospital of Shandong First Medical University & Shandong Provincial Qianfoshan Hospital, Jinan, China, 2 Department of Anesthesiology, Shandong First Medical University, Jinan, China, 3 Department of Nursing, The First Affiliated Hospital of Shandong First Medical University & Shandong Provincial Qianfoshan Hospital, Jinan, China, 4 Department of Anesthesiology, Shandong Provincial Hospital Affiliated to Shandong First Medical University (Shandong Provincial Hospital), Jinan, China

☯ These authors contributed equally to this work.
* ytsun@sdfmu.edu.cn

## Abstract

### Background

To Frailty is associated with postoperative delirium (POD) but is rarely assessed in patients undergoing noncardiac surgery. In this study, the correlation between preoperative frailty and POD, one-year mortality will be investigated in noncardiac Chinese geriatric surgery patients.

### Methods

This study is a prospective, observational, cohort study conducted at a single center with Chinese geriatric patients. Patients who undergo noncardiac surgery and are older than 70 years will be included. A total of 536 noncardiac surgery patients will be recruited from the First Affiliated Hospital of Shandong First Medical University for this study. The Barthel Index (BI) rating will be used to assess the patient's ability to carry out everyday activities on the 1st preoperative day. The modified frailty index (mFI) will be used to assess frailty. Patients in the nonfrailty group will have an mFI < 0.21, and patients in the frailty group will have an mFI ≥ 0.21. The primary outcome is the incidence of POD. Three-Minute Diagnostic Interview for CAM-defined Delirium (3D-CAM) will be conducted twice daily during the 1st-7th postoperative days, or just before discharge. The secondary outcomes will include one-year mortality, in-hospital cardiopulmonary events, infections, acute renal injury, and cerebrovascular events.

relevant data from this study will be made available upon study completion.

**Funding:** 1. Project 2019QL015 supported by the Academic Promotion Programme of Shandong First Medical University. YW, YS, ML, Shandong First Medical University, financial support 20 million RMB, https://www.sdfmu.edu.cn/ . 2. Project 202019170 supported by the Jinan Science and Technology Plan (Clinical Medicine Science and Technology Innovation Plan). YS, MZ, QG, JG, Jinan Science and Technology Bureau, financial support 100 thousand RMB, http://jnsti.jinan.gov. cn/ . 3. Project ZR2022MH221 supported by the Shandong Provincial Natural Science Foundation. YS, MZ, ML, XG, ZG, YG, Natural Science Foundation of Shandong Province, financial support 100 thousand RMB, http://cloud.kjt. shandong.gov.cn/ . The funders had and will not have a role in study design, data collection and analysis, decision to publish, or preparation of the manuscript.

**Competing interests:** The authors have declared that no competing interests exist.

**Abbreviations:** 3D-CAM, 3-Minute Diagnostic Interview for CAM-defined Delirium; aCCI, age-adjusted Charlson Comorbidity Index; BI, Barthel Index; CAM-ICU, confusion assessment model for intensive care unit; CIs, confidence intervals; COPD, chronic obstructive pulmonary disease; CRF, Case Report Form; ICU, intensive care unit; mFI, modified frailty index; MMSE, Mini-Mental State Exam; ORs, odds ratios; PCI, percutaneous coronary intervention; POD, postoperative delirium; TIA, transient ischaemic attack.

## Discussion

This study will clarify the correlation of preoperative frailty with POD and one-year all-cause mortality in Chinese geriatric patients undergoing noncardiac surgery. Can preoperative frailty predict POD or one-year mortality? In the face of China's serious aging social problems, this result may have important clinical value for the surgical treatment of geriatric patients.

## Trial registration

This protocol has been registered with ClinicalTrials. Gov on 12 January 2022 (https:// clinicaltrials.gov/ct2/show/NCT05189678).

## Background

Geriatric adults often suffer from delirium, an acute disorder of attention and cognition, which can be life-threatening. Following an acute illness, surgery, or hospitalization [1], delirium often sets off a series of events that lead to institutionalization, loss of independence, increased morbidity and mortality, and high healthcare expenditure costs [2]. POD is a difficult and complex illness that typically develops 24 to 72 hours after surgery and affects 20% to 80% of geriatric individual patients [3,4]. Aging-related problems are getting worse. More significantly, geriatric patients have undergone surgery frequently over the previous 20 years. Compared to aging, more people are getting sick [5]. In addition, studies have shown that frailty is also more prevalent in the surgical group (frailty percentage 42%–50%) than in the nonsurgical aged population (frailty proportion 4%–10%) [6–8]. Therefore, it is imperative to preoperatively assess the overall health status of geriatric patients and to quickly reverse or lessen their frailty [9].

One of the biggest threats to world health in the twenty-first century is frailty, which goes beyond chronological age. The clinical syndrome of frailty is defined as "a loss of physiological capabilities and reserves in several organ systems that is accompanied by an increased sensitivity to stressors." [10–12]. In spite of the fact that geriatric patients are equally likely to have positive postoperative outcomes as younger patients, fragile patients are more likely to have negative outcomes [13,14]. Postoperative complications can be predicted by preoperative frailty [15,16], including delirium [17,18], falls [19,20], prolonged hospitalizations [21], hospital readmissions [21], discharge to a nursing or assisted-living facility [22], and other surgical complications [16,23]. Elderly frailty is also associated with lower quality of life [8] as well as an accurate predictor of all-cause mortality [16,24,25].

Age, operation type, intensive care unit (ICU) admission, pain, and some drugs are just a few of the many recognized changeable and nonmodifiable risk factors for POD. One of the major risk factors that cannot be changed is age. Accordingly, preoperative screening for geriatric illnesses that are linked to POD and predict poor surgical outcomes is advised for patients undergoing geriatric surgery [26]. The most critical factor is frailty, which is a strong predictor of a range of poor health outcomes in the aged, including falls, disability, and dementia [10,27]. Frailty has been shown to be the most prevalent disease that results in death in geriatric patients [28], which stresses the significance of recognizing frailty in geriatric patients in clinical practice. Frailty is also an independent predictor of postoperative outcome [29]. The therapeutic value of preoperative decision-making and prognostic assessment depends on the

early detection of fragile patients. Preoperative frailty has not yet been proven to be a separate risk factor for POD [30]. This study will look at the relationship between preoperative fragility and the prevalence of POD and one-year mortality undergoing noncardiac surgery.

## Methods/Design

The protocol was written in accordance with the Standard Protocol Items: Recommendations for Interventional Trials guidelines [31]; the protocol is summarised in Figs 1 and 2.

This is a single-centre, prospective, observational, cohort study of preoperative frailty and POD in geriatric patients undergoing noncardiac surgery. Therefore, the study was designed without intervention, randomization and blindness.

### Participants

Patients with an ASA grade I–IV, age ≥ 70 years, and undergoing noncardiac surgery will be chosen from February 2022 to December 2023. The preoperative assessment will make use of the mFI and MMSE. Patients in the nonfrailty group will have a mFI < 0.21, and those in the frailty group will have a mFI ≥ 0.21. Enrollment will end once the projected sample size of patients in both groups has been reached.

### Sample size calculation

The main outcome will be the incidence of POD. Based on literature review and early study results, the estimated incidence of frailty in surgical patients was 23.0%, while the incidence of

| | STUDY PERIOD | | | | | | | | | | | |
|---|---|---|---|---|---|---|---|---|---|---|---|---|
| | Enrolment | Allocation | Post-operation | | | | | | | | Close-out |
| TIMEPOINT | -D1 | 0 | D1 | D2 | D3 | D4 | D5 | D6 | D7 | D30 | one-year |
| **ENROLMENT:** | | | | | | | | | | | |
| **Eligibility screen** | X | | | | | | | | | | |
| **Informed consent** | X | | | | | | | | | | |
| **Allocation** | | X | | | | | | | | | |
| **ASSESSMENTS:** | | | | | | | | | | | |
| **MMSE** | X | | | | | | | | | | |
| **BI** | X | | | | | | | | | | |
| **mFI** | X | X | | | | | | | | | |
| **aCCI** | X | | | | | | | | | | |
| **3D-CAM or CAM-ICU** | | | X | X | X | X | X | X | X | | |
| **Postoperative complications** | | | X | X | X | X | X | X | X | X | |
| **30-day readmission** | | | | | | | | | | X | |
| **one-year mortality** | | | | | | | | | | | X |

**Fig 1. Timeline and schedule for enrolment, allocation, and assessments.** MMSE, Mini-mental State Examination; BI, Barthel Indexl; mFI, modified frailty index; aCCI, age-adjusted Charlson Comorbidity Index; 3D-CAM, 3-Minute Diagnostic Confusion Assessment Method; CAM-ICU, confusion assessment model for intensive care unit.

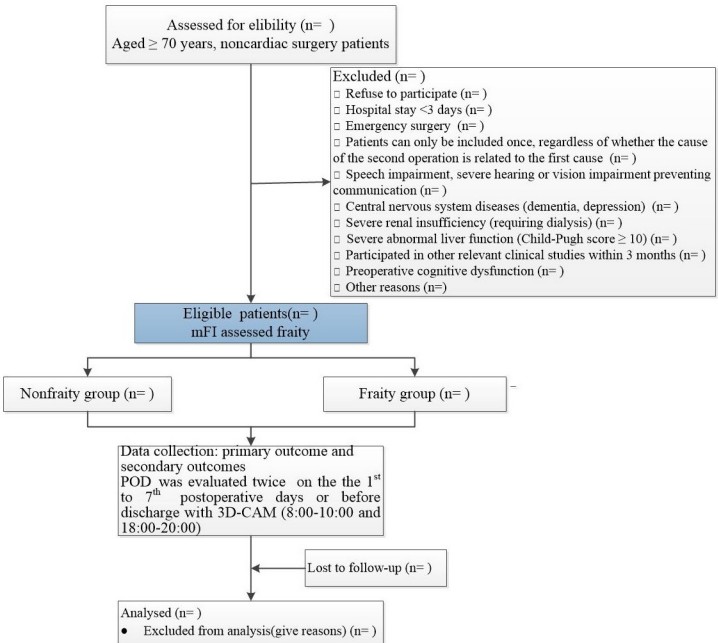

**Fig 2. Flow diagram of the study.**

POD in the frailty and nonfrailty groups was 35.7% and 21.7%. PASS 15.0 software was used to determine the sample size for the independent sample rate comparison of the two groups. The ratio of samples between the experimental and control groups was 0.303, and the two-sided test. Test level (α), and inspection effectiveness (1-β) were set at 0.05 and 0.80, respectively. This calculation yielded a sample size of 112 patients in the frailty group and 370 patients in the nonfrailty group. Considering 10% dropout rate, a total of 536 study volunteers will be required, consisting of 411 patients in the nonfailty group and 125 patients in the failty group.

## Eligibility criteria

Inclusion and exclusion criteria are listed in Table 1.

## Shedding criteria

Study participants who refuse to provide informed consent or withdraw from the study.

## Study implementation

All members of the research team will receive systematic training to master and use the MMSE, aCCI, mFI, BI, 3D-CAM, and CAM-ICU before the study and may only participate after passing the examination.

1. **Preoperative frailty assessment:** The mFI is a National Surgical Quality Improvement Gauge (NSQIP)-based 11-factor index that has been proven to adequately reflect frailty and predict mortality and morbidity [32]. The mFI is calculated by dividing the number of factors present for a patient by the number of available factors for which there are no missing data [33]. An mFI score of 0 is classified as healthy, 0–0.21 is classified as a prefrail, and ≥ 0.21 is classified as a prefrail. The index includes 11 items: nonindependent function

**Table 1. Inclusion and exclusion criteria.**

| Eligibility criteria |
| --- |
| Inclusion criteria |
| 1. Age $\geq$ 70 years; |
| 2. ASA I—IV level; |
| 3. Patients with a signed the informed consent form for the clinical study; |
| 4. Patients scheduled to undergo noncardiac surgery. |
| Exclusion criteria |
| 1. Refuse to participate; |
| 2. Hospital stay <3 days; |
| 3. Emergency surgery; |
| 4. Patients can only be included once, regardless of whether the cause of the second operation is related to the first cause; |
| 5. Speech impairment or severe hearing or vision impairment that prevents communication; |
| 6. Central nervous system diseases (dementia, depression); |
| 7. Severe renal insufficiency (requiring dialysis); |
| 8. Severe abnormal liver function (Child-Pugh score $\geq$ 10); |
| 9. Participation in other relevant clinical studies within 3 months; |
| 10. MMSE examination confirmed the existence of cognitive dysfunction: illiteracy $\leq$ 17 points, primary school level $\leq$ 20 points, secondary school level (including technical secondary school) $\leq$ 22 points, university level (including junior college) $\leq$ 23 points. |

or activity status; history of diabetes; history of chronic obstructive pulmonary disease (COPD) or pneumonia; history of congestive heart failure; history of myocardial infarction; angina pectoris, percutaneous coronary intervention (PCI), cardiac surgery; hypertension requiring medication; peripheral vascular disease or static pain; sensory disturbance; transient ischaemic attack (TIA) attack or cerebrovascular accident without sequelae; cerebrovascular accident with sequelae [34]. In this study, patients with moderate and severe dysfunction with a BI $\leq$ 60 are considered to be positive for item 1. The advantage is that it has been employed and is frequently used in the risk classification of various surgical operations in the NSQIP database of the United States.The majority of evaluation indexes are comorbidities, poor nutrition, metabolism, and other physical function indexes, which is a drawback.

2. **Assessment of activities of daily living:** The patients will be assessed preoperatively by the BI evaluation form; 0–40 is classified as severe dysfunction, 41–60 is classified as moderate dysfunction, 61–99 is classified as mild dysfunction, and 100 is classified as self-care.

3. **Diagnosis of POD:** The 3D-CAM will be used to evaluate POD twice a day (8:00–10:00 and 18:00–20:00) on the the 1st to 7th postoperative days. The 3D-CAM provides a brief assessment (3 orientation items, 4 attention items, 3 symptom probes, and 10 observational items) that facilitates rating of the 4 core CAM features and has a sensitivity of 95% and specificity of 94% when compared with a clinical reference standard rating in a prospective validation study in hospitalized patients [35].

## Primary outcome

The primary outcome will be the incidence of POD.

## Secondary outcomes

The incidence of 30-day readmission, one-year all-cause mortality, postoperative complications (such as pulmonary infection, urinary tract infection, cardiovascular and cerebrovascular accidents, abnormal liver function, postoperative bleeding, incision infection, lower extremity deep venous thrombosis, electrolyte disorder, and hypoproteinemia), and postoperative complications (such as infection of the incision, infection of the incision site, and postoperative bleeding) will all be considered secondary outcomes. According to the attending physicians' definition, a hospital stay is the period of time between admission and discharge.

## Participant timeline

On the day before surgery, candidates will be scrutinized according to the inclusion and exclusion criteria. According to their mFI scores, researchers will classify the participants into the frailty group or the nonfrailty group after receiving written informed consent. We'll keep track of every intraoperative variable.

From the first to the third postoperative day, the patients will be checked on once in the morning and once in the afternoon, and the 3D-CAM score will be taken to gauge the frequency and seriousness of delirium. On the 30th postoperative day, a phone follow-up will be finished, and any issues that develop within 30 days of the procedure will be noted.

## Recruitment

In the day before surgery, we will recruit the patients and explain the study protocol to them. Before deciding whether to participate, the patient will receive enough time to read and assess the information and ask questions. If the patient refuses to participate, the quality of the perioperative management will not be adversely affected.

## Data collection methods

Researchers who are responsible for data management will collect participant data. The data collection files will be standardized to ensure that all data are recorded and can be analyzed in the future. It will be necessary to have at least two researchers (an operator and a research assistant) collect data in each case, and the research assistant will be responsible for supplementing and improving the Case Report Form (CRF) form.

## Data management

According to the findings of the initial observation, researchers will promptly, completely, and accurately enter data on the CRF. According to the plan, the research coordinator will ensure that the study is conducted. The CRF will be delivered to the researchers in charge of data management after being completed and approved by the project's general director. One researcher will enter the data, and a second researcher will review it. CRF will be kept in order. The data management is always subject to review by the ethical committee.

## Patient and public participation

In the creation of research questions, study design, intervention designs, outcome measurements, recruiting, or study execution, patients are not directly involved. The patients will be notified by phone or message of any conference presentations and publications at the conclusion of this study.

## Statistical methods

Continuous variables will be subjected to normality tests (such as age). If the data have a normal distribution, the mean ± standard deviation will be used to represent the data. Intergroup comparisons will be made using independent-sample t tests. In the absence of a normal distribution, medians (interquartile range) will be used to express the data, and the Wilcoxon rank sum test will be applied to compare the groups. A categorical variable, such as the sex or complications, will be expressed as a frequency (percentage). We will use either the chi-square test or Fisher's exact probability approach for intergroup comparisons.

The primary outcomes of POD will be analyzed using a logistic regression model. The correlation between weakness and POD will be calculated using odds ratios (ORs) and 95% confidence intervals (CIs).

Kaplan-Meier survival curves will be used to describe the incidence of secondary outcomes. We will use the log-rank test to compare the differences between groups. A Cox regression model will be used to analyze the influencing factors for 30-day readmissions and complications.

In order to compare the abscission rate between the two groups, the chi-square test will be used.

All statistical analyses will be conducted using the two-sided test with a p value < 0.05 considered statistically significant.

## Ethics approval and consent to participate

Our research is conducted in accordance with the principles of the Declaration of Helsinki (64th WMA General Assembly, October 2013) and was approved by the Ethics Committee of First Affiliated Hospital of Shandong First Medical University. Written informed consent will be obtained from all participants and/or their legal representatives. The results will be disseminated through a peer-reviewed publication and in conferences or congresses.

## Discussion

Older adults with frailty are becoming more prevalent as a result of the rapidly aging population [36,37], which in turn puts more strain on the world's healthcare systems [38]. Older adults with frailty are more likely to experience unmet care needs, fractures and falls, hospitalizations, iatrogenic consequences, and early mortality [14,39–43]. Surgical patients who are geriatric are more likely to experience "age-related events," mainly respiratory complications (pneumonia and respiratory failure), heart problems, postoperative cognitive impairment, an increased likelihood of ICU admission, an extended hospital stay, and higher mortality [44].

Consequently, strategies that target the prevention and management of frailty in an ageing population will probably reduce the burden that the condition imposes on both the individual and health systems. It is recognized that frailty is a clinical syndrome related to aging [45–47] that it is often characterized by a decline in many organ systems' physiological capacity [14,42,47,48], making them more susceptible to stress [14,42,43,45–47]. The functional capacity of a fragile person rapidly diminishes when stressor events occur (such as an acute illness). In order to prevent or reduce frailty from progressing into significant functional impairments, health care policy and supply must implement interventions to prevent or reduce frailty. One of the most typical clinical consequences in geriatric patients is POD [2,3,18]. Its incidence is also directly tied to a number of severe postoperative complications, and the two together are linked to extremely detrimental outcomes. In order to improve perioperative management through early preoperative identification of frail patients, an examination of the impact of preoperative frailty on delirium in geriatric patients undergoing noncardiac surgery is the purpose

of this study. This will help prevent or reduce the occurrence of POD and a number of serious consequences.

According to the inclusion and exclusion criteria, patients will undergo a rigorous screening process. Researchers who have been trained and evaluated will do the preoperative and postoperative follow-up and data collecting for the chosen patients. This research is based on observation. The conclusions will reflect the significance of preoperative frailty assessment and further demonstrate which frailty index indicators are independent risk factors related to POD from the findings of multivariate regression analysis. In geriatric patients undergoing noncardiac surgery, this will serve as a basis for preventing POD.

## Trial status

This is the fourth version of the protocol. The first pre-screened participant was invited to be informed about the study on February 7th, 2022. The recruitment phase or the data collection phase of the trial will be completed by the end of December 2023 including post-test and one-year mortality measurements.

## Supporting information

**S1 File. Study protocol English.**
(DOC)

**S2 File. Study protocol Chinese.**
(DOC)

## Author Contributions

**Conceptualization:** Min Zhang, Hai Feng, Yuelan Wang, Yongtao Sun.

**Data curation:** Min Zhang, Xiaojun Gao, Mengjie Liu, Zhongquan Gao, Xiaxuan Sun, Linlin Huang, Ting Zou, Yongle Guo, Lina Chen, Yang Liu, Xiaoning Zhang, Hai Feng, Yuelan Wang, Yongtao Sun.

**Formal analysis:** Ting Zou.

**Funding acquisition:** Yongtao Sun.

**Investigation:** Min Zhang, Xiaojun Gao, Mengjie Liu, Zhongquan Gao, Xiaxuan Sun, Linlin Huang, Ting Zou, Yongle Guo, Lina Chen, Yang Liu, Xiaoning Zhang, Yuelan Wang, Yongtao Sun.

**Methodology:** Xiaojun Gao, Zhongquan Gao, Xiaxuan Sun, Linlin Huang, Hai Feng, Yuelan Wang, Yongtao Sun.

**Project administration:** Xiaojun Gao, Mengjie Liu, Yang Liu.

**Resources:** Min Zhang, Mengjie Liu, Zhongquan Gao, Xiaxuan Sun, Linlin Huang, Yongle Guo, Xiaoning Zhang.

**Software:** Mengjie Liu, Zhongquan Gao, Lina Chen, Yang Liu.

**Supervision:** Ting Zou, Xiaoning Zhang.

**Validation:** Zhongquan Gao, Xiaxuan Sun, Linlin Huang, Yongle Guo, Yang Liu, Hai Feng, Yuelan Wang.

**Visualization:** Mengjie Liu, Yongle Guo.

**Writing – original draft:** Xiaojun Gao, Yang Liu, Yongtao Sun.

**Writing – review & editing:** Lina Chen, Yongtao Sun.

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
