## [Decision Letter · Decision Letter 0]

3 Feb 2023

PONE-D-22-32948Correlation of preoperative frailty with postoperative delirium and one-year mortality in Chinese geriatric undergoing noncardiac surgery patients: study protocol for a prospective observational cohort studyPLOS ONE

Dear Dr. Sun,

Thank you for submitting your manuscript to PLOS ONE. After careful consideration, we feel that it has merit but does not fully meet PLOS ONE’s publication criteria as it currently stands. Therefore, we invite you to submit a revised version of the manuscript that addresses the points raised during the review process.

We look forward to receiving your revised manuscript.

Kind regards,

Walid Kamal Abdelbasset, Ph.D.

Academic Editor

PLOS ONE

Journal Requirements:

2. Thank you for submitting the above manuscript to PLOS ONE. During our internal evaluation of the manuscript, we found significant text overlap between your submission and previous work in the Methods.

Please revise the manuscript to rephrase the duplicated text, cite your sources, and provide details as to how the current manuscript advances on previous work. Please note that further consideration is dependent on the submission of a manuscript that addresses these concerns about the overlap in text with published work.

We will carefully review your manuscript upon resubmission and further consideration of the manuscript is dependent on the text overlap being addressed in full. Please ensure that your revision is thorough as failure to address the concerns to our satisfaction may result in your submission not being considered further.

3. Please note that PLOS ONE follows the World Health Organization definition of a clinical trial. During the internal evaluation of your article we did not believe your study falls within the scope for clinical trial consideration given the absence of a health intervention. As such please remove all clinical trial documents (ie CT registration number, CONSORT/TREND checklist, CONSORT flowchart, study protocols) and resubmit your article as ‘Research Article’.

"The funders had and will not have a role in study design, data collection and analysis, decision to publish, or preparation of the manuscript."

6. Please amend the manuscript submission data (via Edit Submission) to include author Mengjie Liu.

7. Your ethics statement should only appear in the Methods section of your manuscript. If your ethics statement is written in any section besides the Methods, please move it to the Methods section and delete it from any other section. Please ensure that your ethics statement is included in your manuscript, as the ethics statement entered into the online submission form will not be published alongside your manuscript.

Reviewers' comments:

Reviewer's Responses to Questions

**Comments to the Author**

1. Does the manuscript provide a valid rationale for the proposed study, with clearly identified and justified research questions?

Reviewer #1: Yes

Reviewer #2: Yes

Reviewer #3: Yes

2. Is the protocol technically sound and planned in a manner that will lead to a meaningful outcome and allow testing the stated hypotheses?

Reviewer #1: Yes

Reviewer #2: Yes

Reviewer #3: Yes

3. Is the methodology feasible and described in sufficient detail to allow the work to be replicable?

Reviewer #1: Yes

Reviewer #2: Yes

Reviewer #3: Yes

4. Have the authors described where all data underlying the findings will be made available when the study is complete?

Reviewer #1: Yes

Reviewer #2: Yes

Reviewer #3: Yes

5. Is the manuscript presented in an intelligible fashion and written in standard English?

Reviewer #1: Yes

Reviewer #2: Yes

Reviewer #3: Yes

6. Review Comments to the Author

You may also provide optional suggestions and comments to authors that they might find helpful in planning their study.

Reviewer #1: The authors report a study protocol for assessing the association between preoperative frailty and postoperative delirium and one-year mortality in Chinese patients after noncardiac surgery. Detailed information is reported according to guidelines and SPIRIT checklist has been attached.

It would be better if the authors could clarify following questions:

1. Please clarify the reason why the authors registered patients aged ≥ 70 years, not ≥ 60 or 65 years? In general, the elderly refers to people over the age of 65.

2. Please clarify the participant registration was randomized or not.

3. Please clarify the reason why your study population excluded patients who scheduled to undergo cardiac surgery. Is it due to the modified frailty index includes cardiac surgery? Please further clarify the impact of doing so. This experimental design could affect the generalization of the conclusions. How do you explain that?

4. The modified frailty index was used as diagnostic tools of frailty according to study protocol. Please add related references, and clarify the reason why you used that rather than other criteria such as the Fried's criteria and described the advantages and limitations of that index.

5. The authors stated that “In addition, studies have shown that frailty is also more prevalent in the surgical group (frailty percentage 42%–50%) than in the nonsurgical aged population (frailty proportion 4%–10%) [5-7].” in the introduction, but “Based on literature review and early study results, the estimated incidence of frailty was 23%” in sample size calculation.

The estimated incidence of frailty in your study is more than twice that of non-surgical group in previous studies (and about one-half that of the surgical group). Please clarify the reason why the incidence (23%) was higher than previous studies and possible impact of the high incidence.

Reviewer #2: This is a study protocol to investigate the Correlation of preoperative frailty with postoperative delirium and one-year mortality in

Chinese geriatric undergoing noncardiac surgery patients: study protocol for a

prospective observational cohort study.

At first the title has to be rearranged to :

Correlation of preoperative frailty with postoperative delirium and one-year mortality in Chinese geriatric patients undergoing noncardiac surgery: study protocol for a prospective observational cohort study.

In the abstract: rearrange the sentence: Frailty is associated with postoperative delirium (POD) but is rarely assessed undergoing noncardiac surgery patients

To Frailty is associated with postoperative delirium (POD) but is rarely assessed in patients undergoing noncardiac surgery.

In the Abstract methods: (The study will be carried out in an operating room at a university hospital). This seems strange!

In the Background: (More people are becoming patients than are getting older. )This is grammatically incorrect.

Also, (the connection between preoperative frailty and the prevalence of POD, one-year mortality undergoing noncardiac surgery will be examined in this study), needs correction.

In general, a reasonable study protocol

Reviewer #3: Dear the author of the manuscript. Entitled. Correlation of preoperative reality. With post operative delirium. And one year mortality in Chinese geriatric undergoing noncardiac surgery patients study protocol for a prospective observational cohort study.

Thank you for writing. This protocol for this. Trial. And I congratulate you for the great effort that you are putting in recruiting this number of patients to assess the impact of frailty on postoperative. Delirium after noncardiac surgery.

I believe the protocol is well constructed and well organized. My points are as follows.

1. I guess the status of surgery should be included. In the eligibility criteria. Whether you will include or exclude. Elective., or emergency surgery? I believe the status of surgery will have an impact on. The incidents. Of delirium in the post operative. So, this I guess should be clarified.

2. You mentioned that. A phone follow-up will be finished and any issues that develop within 30 days of the procedure will be noted. I guess in this point you should be clear about. The questions or that you will clarify with the patients because the type of answer depends on the question that you ask. So, is there any way that you mention two or three? Outcomes you will ask about in this follow up.

Otherwise. I have no. Concerns about this protocol. Thank you again for writing this. Protocol.

7. PLOS authors have the option to publish the peer review history of their article (what does this mean?). If published, this will include your full peer review and any attached files.

Reviewer #1: No

Reviewer #2: No

Reviewer #3: **Yes: **salah eldien altarabsheh

---

## [Author Response · Author response to Decision Letter 0]

10 May 2023

Dear Prof. Rose Ann Joyce Sagun Puetes:

Thank you for your letter and for your comments concerning our manuentitled “Correlation of preoperative frailty with postoperative delirium and one-year mortality in Chinese geriatric patients undergoing noncardiac surgery: a prospective observational cohort study” (ID: PONE-D-22-32948R1). Those comments are all valuable and very helpful for revising and improving our paper, as well as the important guiding significance to our researches. We have studied comments carefully and have made correction which we hope meet with approval. Revised portion are marked in red in the paper. 

 The main corrections in the paper and the responds to the reviewer’s comments are as flowing:

Journal Requirements:

We've checked your submission and before we can proceed, we need you to address the following issues:

1. Thank you for submitting your manuscript to PLOS ONE and for responding to our recent requests regarding your submission. Please accept our sincere apologies for the delay in responding to your query. As previously indicated, the CONSORT flow diagram in Figure 2 should be removed. This flow chart is used to show the number of participants excluded/included in the final analysis of a clinical trial. Since your manuscript describes work not yet completed, information on the number of participants included in the analysis cannot be provided. For this reason, it is not appropriate to include a CONSORT diagram in your manuscript. You may include a flow diagram to demonstrate your study design, but a CONSORT diagram is not appropriate for the purpose.

We suggest removing the flow diagram and relying on text in the Methods to describe the study design. Alternatively, you may provide a new diagram to illustrate your study design, but this should be based on the CONSORT diagram.

Response 1 We gratefully appreciate for your valuable suggestion/ comment. We have made changes based on your comments. We have already removed the CONSORT flow diagram. We have modified the "CONSORT flowchart" to the "Flow diagram of the study" and redone the diagram. We don't know if this modification meets the requirements. Can you give us some advice?

Fig. 2 Flow diagram of the study

In addition, one of our coauthors had a change of institution. I have changed the author's information, please know.

Min Zhang1†, Xiaojun Gao1†, Mengjie Liu1, Zhongquan Gao1,3, Xiaxuan Sun1,3, Linlin Huang4, Ting Zou4, Yongle Guo1,3, Lina Chen1, Yang Liu1, Xiaoning Zhang1, Hai Feng1, Yuelan Wang2,3, Yongtao Sun1* 

1 Department of Anesthesiology, The First Affiliated Hospital of Shandong First Medical University & Shandong Provincial Qianfoshan Hospital, Shandong Institute of Anesthesia and Respiratory Critical Medicine, Jinan 250014, China

2 Department of Anesthesiology, Provincial Hospital Affiliated to Shandong First Medical University (Shandong Provincial Hospital), Jinan 250014, China

3 Department of Anesthesiology, Shandong First Medical University, Jinan 250014, China

4 Department of Nursing, The First Affiliated Hospital of Shandong First Medical University & Shandong Provincial Qianfoshan Hospital, Jinan 250014, China

*Corresponding author:

Yongtao Sun, Phone: +8618660795201, Email: ytsun@sdfmu.edu.cn

†These authors also contributed equally to this work.

---

## [Decision Letter · Decision Letter 1]

25 Jul 2023

PONE-D-22-32948R1Correlation of preoperative frailty with postoperative delirium and one-year mortality in Chinese geriatric patients undergoing noncardiac surgery: study protocol for a prospective observational cohort studyPLOS ONE

Dear Dr. Sun,

Thank you for submitting your manuscript to PLOS ONE. After careful consideration, we feel that it has merit but does not fully meet PLOS ONE’s publication criteria as it currently stands. Therefore, we invite you to submit a revised version of the manuscript that addresses the points raised during the review process.

ACADEMIC EDITOR:

I have specific concerns that this is a study protocol of a cohort study, but the authors have included elements of a clinical trial. The authors have included a study flowchart (Figures 2) that is still not suitable as they include numbers of participants or (n= ). This is not appropriate. If you include a flowchart it should only explain your study design. Please revise it accordingly. Alternatively, please remove Figure 2. In addition, in the revised version you do not appear to have provided a response specifically to the original reviewers’ comments. Please ensure that you provide this and a suitably marked up version of the manuscript highlighting how you have responded to all of the points raised (particularly by reviewer #1 in the original review).

We look forward to receiving your revised manuscript.

Kind regards,

Silvia Fiorelli

Academic Editor

PLOS ONE

Journal Requirements:

Reviewers' comments:

Reviewer's Responses to Questions

**Comments to the Author**

1. Does the manuscript provide a valid rationale for the proposed study, with clearly identified and justified research questions?

Reviewer #1: Yes

Reviewer #2: Yes

Reviewer #3: Yes

2. Is the protocol technically sound and planned in a manner that will lead to a meaningful outcome and allow testing the stated hypotheses?

Reviewer #1: Yes

Reviewer #2: Yes

Reviewer #3: Yes

3. Is the methodology feasible and described in sufficient detail to allow the work to be replicable?

Reviewer #1: Yes

Reviewer #2: Yes

Reviewer #3: Yes

4. Have the authors described where all data underlying the findings will be made available when the study is complete?

Reviewer #1: Yes

Reviewer #2: Yes

Reviewer #3: Yes

5. Is the manuscript presented in an intelligible fashion and written in standard English?

Reviewer #1: Yes

Reviewer #2: Yes

Reviewer #3: Yes

6. Review Comments to the Author

You may also provide optional suggestions and comments to authors that they might find helpful in planning their study.

Reviewer #1: I didn't see any response or corrections to my comments. Therefore, I will leave the decision to the editors.

Reviewer #2: I have no comments . This paperwork looks good and I hope that data gathering and analysis will bring new findings

Reviewer #3: Dear the authors of the manuscript entitled "Correlation of preoperative frailty with postoperative delirium and one-year mortality in Chinese geriatric patients undergoing noncardiac surgery: study protocol for a

prospective observational cohort study"

Thank you for taking in consideration all the reviewers comments

I have no concerns about this manuscript

7. PLOS authors have the option to publish the peer review history of their article (what does this mean?). If published, this will include your full peer review and any attached files.

Reviewer #1: No

Reviewer #2: No

Reviewer #3: **Yes: **Salah Eldien Altarabsheh

---

## [Author Response · Author response to Decision Letter 1]

28 Jul 2023

Journal Requirements:

Response 1 We are very sorry for our negligence of. We have made correction according to your comments.

2. Thank you for submitting the above manuscript to PLOS ONE. During our internal evaluation of the manuscript, we found significant text overlap between your submission and previous work in the Methods.

Please revise the manuscript to rephrase the duplicated text, cite your sources, and provide details as to how the current manuscript advances on previous work. Please note that further consideration is dependent on the submission of a manuscript that addresses these concerns about the overlap in text with published work.

We will carefully review your manuscript upon resubmission and further consideration of the manuscript is dependent on the text overlap being addressed in full. Please ensure that your revision is thorough as failure to address the concerns to our satisfaction may result in your submission not being considered further.

Response 2 We are very sorry for our incorrect writing. Using the iThenticate/CrossCheck check tool, we verified our manuscript. We have already made revisions to the paper to remove the redundant content, cite your references, and explain how the present manuscript builds on earlier work. The percentage of text repetition dropped from 36% to 15%.

3. Please note that PLOS ONE follows the World Health Organization definition of a clinical trial. During the internal evaluation of your article we did not believe your study falls within the scope for clinical trial consideration given the absence of a health intervention. As such please remove all clinical trial documents (ie CT registration number, CONSORT/TREND checklist, CONSORT flowchart, study protocols) and resubmit your article as ‘Research Article’.

Response 3, We gratefully appreciate for your valuable suggestion/ comment. I'm sorry, but the cohort study's design was flawed. Under your guidance, we have learned the World Health Organization definition of a clinical trial. Therefore, we revised the manuscript again, studied relevant literature, and modified Figure 2 according to the cohort study. Your suggestions significantly enhanced our post. In this study, the link between preoperative frailty and POD was investigated in a group of Chinese patients aged 70 years or older undergoing noncardiac surgery. The reason for selecting this cohort is that their education level is generally low, which may be different from the world average. The relevance of this study is that we don't have any data on this issue.

1. Braga-Basaria M, Travison TG, Taplin ME, Lin A, Dufour AB, Habtemariam D, Nguyen PL, Kibel AS, Ravi P, Bearup R, Kackley H, Kafel H, Reid K, Storer T, Simonson DC, McDonnell M, Basaria S. Gaining metabolic insight in older men undergoing androgen deprivation therapy for prostate cancer (the ADT & Metabolism Study): Protocol of a longitudinal, observational, cohort study. PLoS One. 2023 Feb 10;18(2):e0281508. doi: 10.1371/journal.pone.0281508. PMID: 36763576.

2. Phillips S, Watt R, Atkinson T, et al. A protocol paper for the MOTION Study-A longitudinal study in a cohort aged 60 years and older to obtain mechanistic knowledge of the role of the gut microbiome during normal healthy ageing in order to develop strategies that will improve lifelong health and wellbeing. PLoS One. 2022;17(11):e0276118. Published 2022 Nov 18. doi:10.1371/journal.pone.0276118

Response 4 We are very sorry for our negligence of ‘Financial Disclosure’. We have matched the the ‘Funding Information’ and ‘Financial Disclosure’ according to your comments, please review.

"The funders had and will not have a role in study design, data collection and analysis, decision to publish, or preparation of the manuscript."

Response 5 Thank you for your rigorous consideration. We have included our amended statements within cover letter and ‘Financial Disclosure’ (via Edit Submission) according to your advice. 

6. Please amend the manuscript submission data (via Edit Submission) to include author Mengjie Liu.

Response 6 Thank you so much for your careful check. We have made correction according to your comments.

7. Your ethics statement should only appear in the Methods section of your manuscript. If your ethics statement is written in any section besides the Methods, please move it to the Methods section and delete it from any other section. Please ensure that your ethics statement is included in your manuscript, as the ethics statement entered into the online submission form will not be published alongside your manuscript.

Response 7 We apologize for the inconvenience this has caused you. The ethics statement has already been relocated to the Methods section, Line 228-234.

Response 8 Thank you for your nice suggestion. On Lines 273-277, we have included Supporting Information.

Review Comments to the Author

Reviewer #1: The authors report a study protocol for assessing the association between preoperative frailty and postoperative delirium and one-year mortality in Chinese patients after noncardiac surgery. Detailed information is reported according to guidelines and SPIRIT checklist has been attached.

It would be better if the authors could clarify following questions:

1. Please clarify the reason why the authors registered patients aged ≥ 70 years, not ≥ 60 or 65 years? In general, the elderly refers to people over the age of 65.

Response 1 We sincerely thank you for your insightful feedback. We had trouble figuring out age when we were planning this study. The age range for senile frailty study outcomes is now 65 or 70 years, with the majority of studies selecting the age range of 70 years. We ultimately settled on 70 years after reading through a substantial amount of literature and evaluating whether the research findings are unambiguous.

1. Morley JE, Vellas B, van Kan GA, et al. Frailty consensus: a call to action. J Am Med Dir Assoc. 2013;14(6):392-397. doi:10.1016/j.jamda.2013.03.022

2. Susano MJ, Grasfield RH, Friese M, et al. Brief Preoperative Screening for Frailty and Cognitive Impairment Predicts Delirium after Spine Surgery. Anesthesiology. 2020;133(6):1184-1191. doi:10.1097/ALN.0000000000003523

3. van Son, Joy E et al. “Atypical presentation of COVID-19 in older patients is associated with frailty but not with adverse outcomes.” European geriatric medicine, 1–11. 7 Feb. 2023, doi:10.1007/s41999-022-00736-z

2. Please clarify the participant registration was randomized or not.

Response 2 Thank you for your nice advice. We have made correction according to your comments, Line 101-102.

3. Please clarify the reason why your study population excluded patients who scheduled to undergo cardiac surgery. Is it due to the modified frailty index includes cardiac surgery? Please further clarify the impact of doing so. This experimental design could affect the generalization of the conclusions. How do you explain that?

Response 3 We gratefully appreciate for your valuable suggestion and understand your concern. Modified frailty indexes included cardiac surgery bias in patients undergoing cardiac surgery. In addition, the incidence of POD in cardiac surgery patients is relatively high, which can reach 50%. Therefore, cardiac surgery and non-cardiac surgery are generally divided into two cohorts for study.

1. Ntalouka, Maria P et al. “The effect of type 2 diabetes mellitus on perioperative neurocognitive disorders in patients undergoing elective noncardiac surgery under general anesthesia. A prospective cohort study.” Journal of anaesthesiology, clinical pharmacology vol. 38,2 (2022): 252-262. doi:10.4103/joacp.JOACP_292_20

2. Bi, Xiaobo et al. “Effects of dexmedetomidine on neurocognitive disturbance after elective non-cardiac surgery in senile patients: a systematic review and meta-analysis.” The Journal of international medical research vol. 49,5 (2021): 3000605211014294. doi:10.1177/03000605211014294

4. The modified frailty index was used as diagnostic tools of frailty according to study protocol. Please add related references, and clarify the reason why you used that rather than other criteria such as the Fried's criteria and described the advantages and limitations of that index.

Response 4 Thank you for your nice suggestion. We have added related references 32 and 33 and made correction according to your comments, Line 145-149.

5. The authors stated that “In addition, studies have shown that frailty is also more prevalent in the surgical group (frailty percentage 42%–50%) than in the nonsurgical aged population (frailty proportion 4%–10%) [5-7].” in the introduction, but “Based on literature review and early study results, the estimated incidence of frailty was 23%” in sample size calculation.

The estimated incidence of frailty in your study is more than twice that of non-surgical group in previous studies (and about one-half that of the surgical group). Please clarify the reason why the incidence (23%) was higher than previous studies and possible impact of the high incidence.

Response 5 We sincerely thank you for your thoughtful advice. We apologize profusely for the surgical patients' negligence. We amended this section since the results of the preliminary study, which were utilized to determine the sample size, were for surgical patients, Line 111. In addition, the difference in incidence rate was due to the type of surgical patients selected. Previous studies did not take non-cardiac surgery patients as research objects, so the incidence rate was as high as 42-50%. In order to ensure the rigor of the trial, our preliminary results showed an incidence of about 23.0% in elderly patients with non-cardiac surgery in China. So there is a difference in incidence in the article.

Reviewer #2: This is a study protocol to investigate the Correlation of preoperative frailty with postoperative delirium and one-year mortality in Chinese geriatric undergoing noncardiac surgery patients: study protocol for a prospective observational cohort study.

At first the title has to be rearranged to:

Correlation of preoperative frailty with postoperative delirium and one-year mortality in Chinese geriatric patients undergoing noncardiac surgery: study protocol for a prospective observational cohort study.

Response 1 Thank you so much for your careful check. We have made correction according to your comments, Line 2.

In the abstract: rearrange the sentence: Frailty is associated with postoperative delirium (POD) but is rarely assessed undergoing noncardiac surgery patients.

To Frailty is associated with postoperative delirium (POD) but is rarely assessed in patients undergoing noncardiac surgery.

Response 2 Thank you for your rigorous consideration. We have made correction according to your comments, Line 20-21.

In the Abstract methods: (The study will be carried out in an operating room at a university hospital). This seems strange!

Response 3 Thank you. We are very sorry for our incorrect writing. We checked the manuscript carefully and found that this sentence was repeated with the following sentence, so it was deleted. 

‘A total of 536 noncardiac surgery patients will be recruited from the First Affiliated Hospital of Shandong First Medical University for this study.’

In the Background: (More people are becoming patients than are getting older.) This is grammatically incorrect.

Response 4 It is really true as your suggested. We have changed the sentence to ‘Compared to aging, more people are getting sick’, Line 57-58. 

Also, (the connection between preoperative frailty and the prevalence of POD, one-year mortality undergoing noncardiac surgery will be examined in this study), needs correction.

Response 5 Thank you. We have re-written this part according to your suggestion, Line 86-88. 

In general, a reasonable study protocol

Reviewer #3: Dear the author of the manuscript. Entitled. Correlation of preoperative reality. With post operative delirium. And one year mortality in Chinese geriatric undergoing noncardiac surgery patients study protocol for a prospective observational cohort study.

Thank you for writing. This protocol for this. Trial. And I congratulate you for the great effort that you are putting in recruiting this number of patients to assess the impact of frailty on postoperative. Delirium after noncardiac surgery.

I believe the protocol is well constructed and well organized. My points are as follows.

1. I guess the status of surgery should be included. In the eligibility criteria. Whether you will include or exclude. Elective., or emergency surgery? I believe the status of surgery will have an impact on. The incidents. Of delirium in the post operative. So, this I guess should be clarified.

Response 1 We gratefully appreciate for your valuable suggestion. We have added patients receiving elective non-cardiac surgery to clause 4 of the inclusion criteria. Moreover, add a third urgent surgery to the list of disqualifying factors. The design of this study selects non-cardiac surgery patients in the cohort, without distinguishing specific surgical methods. In the event of postoperative delirium, CAM-ICU will be used for assessment. With your professional advice, we have added CAM-ICU, Line33-35. Once again, I want to thank you for your expert guidance, which raised the article's level of excellence.

2. You mentioned that. A phone follow-up will be finished and any issues that develop within 30 days of the procedure will be noted. I guess in this point you should be clear about. The questions or that you will clarify with the patients because the type of answer depends on the question that you ask. So, is there any way that you mention two or three? Outcomes you will ask about in this follow up.

Response 2 Thank you for your valuable suggestion. Telephone interviews conducted 30 days after surgery asked about all-cause deaths, readmissions, repeat surgeries, tumor recurrences, cardiac arrest, etc.

Otherwise. I have no. Concerns about this protocol. Thank you again for writing this. Protocol.

We tried our best to improve the manu and made some changes in the manu. These changes will not influence the content and framework of the paper. And here we did not list the changes but marked in red in revised paper. We appreciate for Editors/Reviewers’ warm work earnestly, and hope that the

correction will meet with approval. Once again, thank you very much for your comments and suggestions.

---

## [Decision Letter · Decision Letter 2]

6 Sep 2023

PONE-D-22-32948R2Correlation of preoperative frailty with postoperative delirium and one-year mortality in Chinese geriatric patients undergoing noncardiac surgery: study protocol for a prospective observational cohort studyPLOS ONE

Dear Dr. Sun,

Thank you for submitting your manuscript to PLOS ONE. After careful consideration, we feel that it has merit but does not fully meet PLOS ONE’s publication criteria as it currently stands. Therefore, we invite you to submit a revised version of the manuscript that addresses the points raised during the review process.

ACADEMIC EDITOR:Thank you for carefully assess all reviewers points.I still have a specific comment on figure 2. 

Since your manuscript describes work not yet completed, information on the number of

participants included in the analysis cannot be provided.

so you should remove all the patients numbers in this figure and leave " (n=)"

thank you

We look forward to receiving your revised manuscript.

Kind regards,

Silvia Fiorelli

Academic Editor

PLOS ONE

Journal Requirements:

Reviewers' comments:

Reviewer's Responses to Questions

**Comments to the Author**

1. Does the manuscript provide a valid rationale for the proposed study, with clearly identified and justified research questions?

Reviewer #1: Yes

Reviewer #3: Yes

2. Is the protocol technically sound and planned in a manner that will lead to a meaningful outcome and allow testing the stated hypotheses?

Reviewer #1: Yes

Reviewer #3: Yes

3. Is the methodology feasible and described in sufficient detail to allow the work to be replicable?

Reviewer #1: Yes

Reviewer #3: Yes

4. Have the authors described where all data underlying the findings will be made available when the study is complete?

Reviewer #1: Yes

Reviewer #3: Yes

5. Is the manuscript presented in an intelligible fashion and written in standard English?

Reviewer #1: Yes

Reviewer #3: Yes

6. Review Comments to the Author

You may also provide optional suggestions and comments to authors that they might find helpful in planning their study.

Reviewer #1: The authors have responded to the comments point by point. I have no further comments.

(The uploaded manuscripts seem to be a bit confusing, there are three manuscripts in the pdf. Whether Fig2 was deleted or revised in the final version? and I saw that the editor suggested that the manuscript should be submitted as a Research Article, but R2 revision seems to be still a Study Protocol.)

Reviewer #3: Dear the authors

I read the revised version of the manuscript and your responses to the reviewers' comments.

I believe this manuscript in its current version stands in solid shape.

I have no concerns

7. PLOS authors have the option to publish the peer review history of their article (what does this mean?). If published, this will include your full peer review and any attached files.

Reviewer #1: No

Reviewer #3: **Yes: **salah eldien altarabsheh

---

## [Author Response · Author response to Decision Letter 2]

8 Sep 2023

Dear Academic Editor Silvia Fiorelli:

Thank you for your letter and for your comments concerning our manuentitled “Correlation of preoperative frailty with postoperative delirium and one-year mortality in Chinese geriatric patients undergoing noncardiac surgery: a prospective observational cohort study” (ID: PONE-D-22-32948R1). Those comments are all valuable and very helpful for revising and improving our paper, as well as the important guiding significance to our researches. 

ACADEMIC EDITOR:

Thank you for carefully assess all reviewers points.

I still have a specific comment on figure 2. 

Since your manuscript describes work not yet completed, information on the number of

participants included in the analysis cannot be provided.

so you should remove all the patients numbers in this figure and leave " (n=)". 

Response 1: Thank you for the constructive comments and suggestions. We have removed remove all the patients numbers in this figure and leave " (n=)".

---

## [Decision Letter · Decision Letter 3]

24 Nov 2023

Correlation of preoperative frailty with postoperative delirium and one-year mortality in Chinese geriatric patients undergoing noncardiac surgery: study protocol for a prospective observational cohort study

PONE-D-22-32948R3

Dear Dr. Sun,

We’re pleased to inform you that your manuscript has been judged scientifically suitable for publication and will be formally accepted for publication once it meets all outstanding technical requirements.

Kind regards,

Silvia Fiorelli

Academic Editor

PLOS ONE

Additional Editor Comments (optional):

Reviewers' comments:

Reviewer's Responses to Questions

**Comments to the Author**

1. Does the manuscript provide a valid rationale for the proposed study, with clearly identified and justified research questions?

Reviewer #3: Yes

2. Is the protocol technically sound and planned in a manner that will lead to a meaningful outcome and allow testing the stated hypotheses?

Reviewer #3: Yes

3. Is the methodology feasible and described in sufficient detail to allow the work to be replicable?

Reviewer #3: Yes

4. Have the authors described where all data underlying the findings will be made available when the study is complete?

Reviewer #3: Yes

5. Is the manuscript presented in an intelligible fashion and written in standard English?

Reviewer #3: Yes

6. Review Comments to the Author

You may also provide optional suggestions and comments to authors that they might find helpful in planning their study.

Reviewer #3: I am satisfied with the manuscript in its current version

Thnak you for considering all the reviewers comments

7. PLOS authors have the option to publish the peer review history of their article (what does this mean?). If published, this will include your full peer review and any attached files.

Reviewer #3: **Yes: **salah Eldien altarabsheh

---

## [Editor Report · Acceptance letter]

25 Feb 2024

PONE-D-22-32948R3 

PLOS ONE

Dear Dr. Sun, 

I'm pleased to inform you that your manuscript has been deemed suitable for publication in PLOS ONE. Congratulations! Your manuscript is now being handed over to our production team.

Kind regards, 

on behalf of

Dr. Silvia Fiorelli 

Academic Editor

PLOS ONE